# The side effect profile of Clozapine in real world data of three large mental health hospitals

Ehtesham Iqbal[1]*, Risha Govind[1], Alvin Romero[2], Olubanke Dzahini[3], Matthew Broadbent[4,5], Robert Stewart[4,5,6], Tanya Smith[7,8], Chi-Hun Kim[9], Nomi Werbeloff[10,11], James H. MacCabe[4,12], Richard J. B. Dobson[1,4,5,13,14], Zina M. Ibrahim[1,4,5,13,14]*

1 The Department of Biostatistics and Health Informatics, Institute of Psychiatry, Psychology and Neuroscience, King's College London, London, United Kingdom, 2 SLAM BioResource for Mental Health, South London and Maudsley NHS Foundation Trust and King's College London, London, United Kingdom, 3 Pharmacy Department, South London and Maudsley NHS Foundation Trust, London, United Kingdom, 4 NIHR Biomedical Research Centre for Mental Health, South London and Maudsley NHS Foundation, London, United Kingdom, 5 Biomedical Research Unit for Dementia, South London and Maudsley NHS Foundation, London, United Kingdom, 6 Department of Health Service & Population Research, Institute of Psychiatry, Psychology and Neuroscience, King's College London, London, United Kingdom, 7 Oxford Health NHS Foundation Trust, Oxford, United Kingdom, 8 NIHR Oxford Health Biomedical Research Centre, University of Oxford and Oxford Health NHS Foundation Trust, Oxford, United Kingdom, 9 Department of Psychiatry, University of Oxford, Oxford, United Kingdom, 10 UCL Division of Psychiatry, University College London, London, United Kingdom, 11 Camden and Islington, NHS Foundation Trust, London, United Kingdom, 12 Psychosis Studies, Institute of Psychiatry, Psychology and Neuroscience, King's College London, De Crespigny Park, London, United Kingdom, 13 The Farr Institute of Health Informatics Research, UCL Institute of Health Informatics, University College London, London, United Kingdom, 14 NIHR Biomedical Research Centre, University College London Hospitals, London, United Kingdom

* Ehtesham.iqbal@kcl.ac.uk (EI); zina.ibrahim@kcl.ac.uk (ZMI)

## Abstract

### Objective

Mining the data contained within Electronic Health Records (EHRs) can potentially generate a greater understanding of medication effects in the real world, complementing what we know from Randomised control trials (RCTs). We Propose a text mining approach to detect adverse events and medication episodes from the clinical text to enhance our understanding of adverse effects related to Clozapine, the most effective antipsychotic drug for the management of treatment-resistant schizophrenia, but underutilised due to concerns over its side effects.

### Material and methods

We used data from de-identified EHRs of three mental health trusts in the UK (>50 million documents, over 500,000 patients, 2835 of which were prescribed Clozapine). We explored the prevalence of 33 adverse effects by age, gender, ethnicity, smoking status and admission type three months before and after the patients started Clozapine treatment. Where possible, we compared the prevalence of adverse effects with those reported in the Side Effects Resource (SIDER).

**Data Availability Statement:** SLAM: The ethical approval to access CRIS data requires the data to be stored behind the hospital's firewall with access monitored by strict guidelines via a patient-led

oversight and governance committee. It is due to this restriction that the data cannot be made available within the manuscript, supporting files or via a public repository. However, data access for research purposes is possible to subject to approval from the oversight committee and can be initiated by contacting the CRIS oversight committee (cris.administrator@SLAM.nhs.uk). Camden & Islington: The data used in this work has been obtained from the Clinical Record Interactive Search (CRIS), a system which has been implemented at the Camden & Islington NHS Foundation Trust (C&I). It provides authorised researchers with regulated access to anonymised information extracted from patient electronic health records. CRIS is governed by a strict information governance scheme which forbids anyone except for authorised researchers from accessing its records. Access to CRIS is restricted to 1) C&I employees or 2) those having an honorary contract or letter of access from the Trust. Once an honorary contract is established, researchers can only access CRIS once they submit a research project proposal through the CRIS Project Application form. The form is available here: http://www.candi.nhs.uk/health-professionals/research/ci-research-database/researchers-and-clinicians For further details, contact: researchdatabase@candi.nhs.uk. Oxford: CRIS provides a means of analysing anonymised data from the Oxford Health NHS Foundation Trust electronic case records. Ethical approval for such analyses was provided by Oxfordshire REC C National Research Ethics Service in July 2015. Access to clinical information is clearly a sensitive issue, and a security model was developed which has been considered and approved by the Oxford Health NHS Foundation Trust Caldicott Guardian and the Trust Executive, as well as forming part of the ethics application.

**Funding:** This paper represents independent research funded by the National Institute for Health Research (NIHR) Biomedical Research Centre at South London and Maudsley NHS Foundation Trust and King's College London, and by the NIHR Oxford Health Biomedical Research Centre at Oxford University and Oxford Health NHS Foundation Trust (grant BRC-1215-20005). This research was supported by researchers at the National Institute for Health Research University College London Hospitals Biomedical Research Centre, and by awards establishing the Farr Institute of Health Informatics Research at UCL Partners, from the Medical Research Council, Arthritis Research UK, British Heart Foundation, Cancer Research UK, Chief Scientist Office, Economic and Social Research Council,

## Results

Sedation, fatigue, agitation, dizziness, hypersalivation, weight gain, tachycardia, headache, constipation and confusion were amongst the highest recorded Clozapine adverse effect in the three months following the start of treatment. Higher percentages of all adverse effects were found in the first month of Clozapine therapy. Using a significance level of ($p < 0.05$) our chi-square tests show a significant association between most of the ADRs and smoking status and hospital admission, and some in gender, ethnicity and age groups in all trusts hospitals. Later we combined the data from the three trusts hospitals to estimate the average effect of ADRs in each monthly interval. In gender and ethnicity, the results show significant association in 7 out of 33 ADRs, smoking status shows significant association in 21 out of 33 ADRs and hospital admission shows the significant association in 30 out of 33 ADRs.

## Conclusion

A better understanding of how drugs work in the real world can complement clinical trials.

## Introduction

Adverse Drugs Reactions (ADRs) are troublesome, potentially life-threatening, and associated with more extended hospital stays. Improving our ability to detect these unwanted effects can lead to a greater understanding of individual risks and better patient management, which would help reduce the burden on healthcare providers [1–3]. Clinical trials provide the first insight into ADR profiles before a drug is available to the public. Once the drug is on the market, new ADRs and medication errors are spontaneously reported by physicians and pharmacists through a number of schemes (e.g. the YellowCard scheme used by the UK Medicine Health Regulatory Agency, MHRA, and the Food and Drug Administration (FDA) Adverse Event Reporting System (FAERS) in the US [4]). However, these spontaneous reporting systems have limitations. There can be under-reporting of observed ADRs, as well as no reporting of ADRs that are perceived as non-serious. It is possible that many novel ADRs are never entered into the system because it is challenging to establish a causal link between the drug and adverse events. The data collected in Electronic Health Records (EHRs) as part of routine care may provide additional insight into adverse events in real-world settings as well as potentially identifying new positive indications of drugs [5].

Clozapine is a atypical, also known as second-generation, antipsychotic drug. It is widely recognised as the gold standard in the treatment of schizophrenia and the most effective antipsychotic drug in the management of treatment-resistant schizophrenia [6, 7]. Nevertheless, Clozapine is an underutilised medication [8] with only 54% of all eligible patients being prescribed Clozapine in the UK [9]. One of the primary reasons for its limited use is the concern over its side effects, some of which are potentially fatal and require frequent monitoring [10]. Two common side effects associated with Clozapine are weight gain and hypoglycaemia, which together can lead to type II diabetes [11, 12]. Others include abdominal pain, agitation, akathisia, amnesia, blurred vision, confusion, constipation, convulsions, delirium, delusion, diarrhoea, dizziness, dry mouth, enuresis, fatigue, fever, headache, heartburn, hallucination, hyperkinesia, hypersalivation, hypertension, hypotension, insomnia, nausea, rash, restlessness, seizures, sleeplessness, sweating, syncope, tachycardia, tremor, vomiting and decrease in White blood cells (WBC) [13–16]. These side effects can be dose-related [17, 18]. A more acute side effect is agranulocytosis, a severe and dangerous lowered WBC count. Therefore, as

Engineering and Physical Sciences Research Council, National Institute for Health Research, National Institute for Social Care and Health Research, and Wellcome Trust (grant MR/K006584/1). This study was supported by the Case Record Interactive Search (CRIS) system funded and developed by the National Institute for Health Research (NIHR) Mental Health Biomedical Research Centre at South London and Maudsley NHS Foundation Trust and King's College London using the system within the NIHR Oxford Health Biomedical Research Centre (Oxford University and Oxford Health NHS Foundation Trust).

**Competing interests:** The authors have declared that no competing interests exist.

soon as the patient starts taking Clozapine, blood monitoring begins to determine whether or not the patient is at risk of agranulocytosis [10, 19–21]. Other severe but rare side effects include myocarditis, neutropenia, cardiomyopathy, Creatinine Phosphokinase (CPK) increase, hepatic narcosis and Steven Johnson Syndrome (SJS) [22].

Several studies have report Clozapine-induced ADRs, but those are generally limited by the duration of the study (6 weeks to 2 years), cohort size (31 to 110 patients) and the number of the ADRs discussed (1 to 18) [23–28]. Limited work has been done to characterise a large population of patients experiencing Clozapine-induced ADRs.

In addition, little work has been done to understand the relationship between age and Clozapine-induced ADRs. The literature suggests that there is a positive relationship between age and Clozapine-induced ADRs such as weight gain and cardiovascular risks, mainly myocarditis and cardiomyopathy [29, 30]. In young patients, several short-term and long-term studies have reported Clozapine-induced ADRs in different hospital settings. However, those only cover a single or a few ADRs at a time, rather than a range of ADRs [31–39].

Schizophrenia patients are more likely to be smokers as compared to patients with other psychiatric disorders [40]. They are more likely to be heavy smokers (25 or more cigarettes daily) compared with only 11% of the general population of smokers [41, 42]. The chemicals in cigarette smoke induce enzymes that accelerate the metabolism of antipsychotic drugs [43, 44], requiring a higher dosage of Clozapine in smokers compared to non-smokers. There is no study showing comprehensive profiling of Clozapine-induced ADRs in the smoker vs non-smoker patient populations.

Males and Females exhibit different responses to drug treatment [45, 46]. Studies suggest that women are at a higher risk of ADRs than men, and ADR-related hospital admissions are more common in females and older patients [47, 48]. Some studies suggest that weight gain, hypertension, dyslipidemia and other metabolic synderomes are more prevalent in female Clozapine patients, whereas cardiac-related ADRs such as arrhythmia, QT prolongation are more prevalent in males [49, 50].

Although often ignored, ethnicity plays a significant role in response to psychotropic medication [51, 52]. Studies suggest Clozapine is used much less in the black population compared to any other ethnicity due to the lower normal range of WBC count [52, 53]. Clozapine is also known to cause Type II diabetes in the black and Asian population [12, 54, 55].

We developed a natural language processing (NLP) pipeline, ADEPt [56] to detect and validate mentions of adverse drug events in free-text mental health clinical notes. The ADEPt is equipped with an improved rule-based, ADE dictionary and modified ConText algorithm and distinguish from positive to negative mentions of ADEs, from current to past ADEs occurrences by applying temporal reasoning and from general to patient-specific ADEs mentions.

In this paper, we combine information about adverse drug events using the ADEPt pipeline with medication episodes mined from clinical text to enhance the understanding of adverse effects related to Clozapine. We also characterise the population experiencing ADRs with Clozapine and compare the results with the Side Effects Resource (SIDER) [57], complementing what is already known about demographics, smoking status and hospital admissions, and to find out how different subgroups are most likely to experience ADRs when administered Clozapine.

## Material and methods

### Data sources

We obtained data from the Clinical Record Interactive Search (CRIS) system, a de-identified version of the EHR of three large mental health National Health Services (NHS) Foundation

Trusts in the UK, made available for research through a patient-led model of governance and oversight [58]: i) the South London and Maudsley (SLAM) NHS Foundation Trust is one of the largest mental health care providers in Europe serving a population over 1.4 million residents in four major hospitals London [59], ii) the Camden and Islington (C&I) NHS Foundation Trust provides mental health services in North London [60, 61] and iii) Oxford Health NHS Foundation Trust, which covers a geographic catchment area containing 1.9 million UK residents [62]. The Trust is a group or network of hospitals that work together to deliver a range of services to their catchment area. Both SLAM and C&I trusts represent large, mixed, multicultural areas of London, UK.

SLAM data came from patients who received care from January 2007 to December 2016; Camden & Islington data came from patients who received care from June 2009 to December 2015, and Oxford data came from patients who received care from January 2010 to December 2014.

The combined data of the three trusts comprises over 500,000 patient records and over 50 million documents. We used both the unstructured and structured data within the EHRs to identify periods of on-off Clozapine medication, and then to identify subsequent Clozapine-induced ADRs.

## Mining medication start and stop dates

Using the General Architecture for Text Engineering (GATE) medication NLP application [59], a Natural Language processing system that has been repeatedly used in drug-related studies [59, 63, 64], we developed an algorithm to infer periods of 'on-treatment' medication episodes.

We created a drug dictionary using 260 drugs commonly used in the psychiatric hospitals for Mental, Behavioural and Neurodevelopmental Disorders. The drug dictionary consists of 11 drug categories: Antidepressants, Antidiabetics, Antiepileptics, Antihypertensives, Antipsychotics, Anti-Dementia, Hypnotics & Anxiolytics, Lipid Regulatory, Mood Stabilizers, Non-Steroidal Anti-Inflammatory and Anti-Parkinson. We used the British National Formulary (BNF) and electronic Medicines Compendium (eMC) [65, 66] to generate a robust list of all available brand names. In order to increase coverage, we included brand names that have been discontinued, as these may be present in the clinical notes. The algorithm has been previously used in a dementia study to identify trajectories of cognitive decline in the SLAM EHR [67].

The algorithm works in two stages. In the first stage, it maps all brand names onto the generic names of the drugs. In the second stage, the algorithm sorts the records by date and measures the length of the interval between consecutive dates. These dates include prescription dates and positive mentions of a drug in the clinical notes, indicating that a patient is taking/continuing a particular drug. We extracted dated of positive mentions of medications using a specialised GATE NLP application [59, 63, 64].

We set the threshold for determining a medication episode between two consecutive prescribing dates to 42 days (6 weeks). Although the common practice for prescribing psychotropic drugs is 28 days (4 weeks), they can be prescribed for 42 days based on patient availability. If the gap between two consecutive dates is less than 42 days (6 weeks), the algorithm searches for the next date until it finds the date where the difference between two dates is greater than 42 days, or no data point is available. In each episode, the algorithm counts the number of data points it used to conclude an episode. Once the difference between two dates is over 42 days, the algorithm concludes the episode using the last available date, and subsequently starts a new episode. The duration threshold can be changed according to the prescribing practices associated with the medication being studied.

## Mining adverse events from clinical text

We used our Adverse Drug Event annotation Pipeline ADEPt to mine ADEs from the free-text data. We applied the pipeline to extract 66 ADEs on the Oxford data and 110 ADEs from the SLAM and C&I data (more recent work used a larger ADE dictionary).

## Associations between medications & ADRs: Formulating an ADR timeline

We set out to uncover associations between medication episodes and adverse events by combining the medication timeline with the adverse events mined by the ADEPt pipeline. This is done in a number of steps: First, multiple discussions of an ADE on a given day are collapsed into a single event, filtering out negative mentions accordingly. The ADR algorithm then queries the medication timeline to identify drugs that the patient was taking at the time of an adverse event and then creates an ADR event.

## Statistical analysis

The study used the chi-square statistical test with Bonferroni correction to quantify the significance of ADR associations in relation to gender, ethnic background, age, and smoking and hospital status. The data taken into account was based on a monthly interval after starting the drug clozapine. R programming language version 3.2.4 was used to conduct statistical analysis.

## Clozapine cohort and associated variables

Once periods of prescribing and associated possible ADRs have been identified, the prevalence of ADEs across different subpopulations within the EHR was explored. As such, age, gender, ethnicity, smoking status, hospital admissions (inpatient/outpatient) were extracted for each patient in each Trust, where data was available. Although efforts were taken to replicate the study in all three NHS Trusts, the identification and extraction of age groups, smoking status and inpatient/outpatient status in Oxford NHS Trust data was not successful.

   We retrieved the date of birth, gender, ethnicity, smoking status, hospital admission and diagnosis from each trust. A patient's age was calculated using the date the patients started Clozapine and was further divided into eight distinct categories. Gender and ethnicity were derived from the latest entry recorded in each NHS trust. The ethnicity was divided into four major groups, white, black, Asians and others. Smoking status was calculated six months before and after the date that the patient started Clozapine. Hospital admission status was measured by looking into the patient admission and discharge data. If the patient started the Clozapine during their hospital admission, we classified them as an inpatient.

   We established six categories of diagnoses according to using ICD-10 codes. Three of these categories came from Serious Mental Illness (SMI), Schizophrenia (ICD-10: F20-F29) excluding (ICD-10: F25), Schizoaffective (ICD-10: F25) and Bipolar (ICD-10: F31). The other three categories were any mental, behavioural and neurodevelopmental disorders (ICD-10: F01-F99) excluding SMI patients, any other diagnosis and diagnosis not available. We collected diagnoses six months before and after the first prescription of Clozapine. Where we were unable to identify records of diagnosis during a year period, we increased the time span until a diagnosis was established for a patient. We compared our results to SIDER where possible, to understand the prevalence of Clozapine-related ADRs in the real-world EHR data we have collected.

## Results

We identified 2835 patients who have taken Clozapine for at least three months in three large mental health trusts. Table 1 gives the cohort characteristics of the three trusts with respect to gender, ethnic background, age groups, smoking status, hospital admission and diagnosis.

The ADR algorithm was implemented across the three NHS trusts and was evaluated in SLAM and Camden & Islington to assess its performance in detecting associations between drugs and ADEs; access limitations prohibited evaluation in the Oxford trust. A set of 300 cases were randomly selected from each trust, and manual validation was carried out by two annotators in SLAM and one annotator in C&I reading through the clinical notes according to the following criteria:

a. The presence of ADE is a true positive.

b. The associated medication episode annotation was a true positive.

The algorithm achieved a 0.89 Positive Predictive Value (PPV) in SLAM and a 0.87 PPV in C&I. The False Discovery Rate (FDR) was 0.1 in SLAM and 0.12 in C&I.

In SLAM, the level of agreement between the two annotators for ADEs 93% and kappa (κ = 0.56) and for drugs 94% and kappa (κ = 0.56) was achieved.

SIDER contains side effects information from RCTs and systems such as the FAERS [4]. As of January 2018, SIDER contained 23 different sources on Clozapine adverse effects such as post-marketing, FDA, labels, Medsafe and Health Canada. Although SIDER collates data from a number of sources, much of its data is from RCTs, which are typically run on small and narrow populations, with little information on how medications work in real-world settings. The study results were compared with SIDER where possible; to understand the prevalence of Clozapine-induced ADRs in the real-world EHR data.

The comparison results with SIDER are summarised in Table 2 for 33 ADRs. The results show the percentages of patients by ADR in each trust (SLAM, Camden & Islington and Oxford) and SIDER. The results are stratified into monthly intervals from the initiation of Clozapine treatment, three months prospective and three months retrospective. The columns (Three Months Early, Two Months Early, One Month Early, One Month Later, Two Months Later, and Three Months Later) show the percentages in each monthly interval. The last two columns (SIDER Low End and SIDER High End) show the SIDER reporting from different clinical trials and FDA studies. The complete results stratified into monthly intervals are provided in the (S1 Table) for gender, ethnicity, age groups, smoking status and hospital admissions.

The statistical analysis was first carried out separately for each of the trust. To prevent influence of cofounder such as age, gender, ethnicity, smoking status and hospital admission were performed individually.

The results are available in (S2 Table). The p-value of the test is adjusted through Bonferroni correction. Gender and ethnicity showed no significant association with any ADRs. Age group showed significant association with agitation, fatigue, feeling sick, sedation and tachycardia. In hospital admission, of the 33 ADRs, 15 were found to have a significant association (abdominal pain, agitation, confusion, dizziness, diarrhoea, fatigue, headache, hypersalivation, hypotension, hypertension, insomnia, sedation, tachycardia, tremor and vomiting). Smoking status showed that of the 33 ADRs, 8 were found to have a significant association (abdominal pain, agitation, confusion, dizziness, diarrhoea, fatigue, sedation and tachycardia).

The datasets from the three Trusts were later combined, and the chi-square statistical test was performed to estimate the average effect of ADRs in each monthly interval. The combined analysis showed a significant frequency distribution after Bonferroni p-value adjustment in

**Table 1. Cohort characteristics of the three NHS trusts, showing a breakdown of gender, ethnic background, age groups, smoking status, hospital admission and diagnosis.**

| Cohort | SLAM | Camden & Islington | Oxford | Total |
|---|---|---|---|---|
| **Size** | 1760 | 561 | 514 | 2835 |
| **Gender** | | | | |
| **Male** | 1167 | 357 | 342 | 1866 |
| | 66.3% | 63.6% | 66.5% | |
| **Female** | 593 | 204 | 172 | 969 |
| | 33.7% | 36.3% | 33.4% | |
| **Ethnic Background** | | | | |
| **White** | 821 | 347 | 426 | 1594 |
| | 46% | 62% | 83% | |
| **Black** | 704 | 120 | 20 | 844 |
| | 40% | 21.39% | 4% | |
| **Asian** | 93 | 41 | 41 | 175 |
| | 5.3% | 7.31% | 8% | |
| **Other** | 142 | 53 | 27 | 222 |
| | 8% | 9.45% | 5.25% | |
| **Age Group** | | | | |
| **Under 21** | 57 | | 12 | 69 |
| | 3.2% | | 2.33% | |
| **21–30** | 422 | 27 | 96 | 545 |
| | 24% | 4.81% | 18.68% | |
| **31–40** | 488 | 141 | 155 | 784 |
| | 28% | 25.13% | 30.16% | |
| **41–50** | 479 | 168 | 126 | 773 |
| | 27% | 30% | 24.51% | |
| **51–60** | 233 | 135 | 98 | 466 |
| | 13% | 24% | 19% | |
| **61–70** | 62 | 67 | 22 | 151 |
| | 3.5% | 12% | 4.28% | |
| **71–80** | 18 | 21 | 4 | 43 |
| | 1% | 3.74% | 0.78% | |
| **Above 80** | 1 | 2 | 1 | 4 |
| | 0.06% | 0.35% | 0.19% | |
| **Smoking Status** | | | | |
| **Smoker** | 1039 | 360 | | 1399 |
| | 59% | 64% | | |
| **Non-Smoker** | 721 | 201 | | 922 |
| | 41% | 36% | | |
| **Hospital Admission** | | | | |
| **Inpatient** | 737 | 114 | | 851 |
| | 42% | 20% | | |
| **Outpatient** | 1023 | 447 | | 1470 |
| | 58% | 80% | | |
| **Diagnosis** | | | | |
| **Schizophrenia (ICD-10: F20-F29) excluding (ICD-10: F25)** | 1355 | 411 | 356 | 2122 |
| | 77% | 73% | 69% | 75% |

(*Continued*)

**Table 1.** (Continued)

| Cohort | SLAM | Camden & Islington | Oxford | Total |
|---|---|---|---|---|
| **Schizoaffective (ICD-10: F25)** | 260 | 45 | 61 | 366 |
| | 15% | 8% | 12% | 13% |
| **Bipolar (ICD-10: F31)** | 44 | 26 | 11 | 81 |
| | 3% | 5% | 2% | 3% |
| **Any mental, behavioural and neurodevelopmental disorders (ICD-10: F01-F99) excluding SMI patients** | 32 | 17 | 11 | 60 |
| | 2% | 3% | 2% | 2% |
| **Any other Diagnosis** | 54 | 42 | 39 | 135 |
| **Excluding (ICD-10: F01-F99)** | 2% | 7% | 8% | 5% |
| **Diagnosis** | 15 | 21 | 36 | 72 |
| **Not Available** | 1% | 4% | 7% | 2% |

the categorical variables (gender, ethnicity, age group, hospital admissions and smoking status) and a number of the ADRs as follows:

a. Gender showed significant associations (See Fig 1) with backache, constipation, diarrhoea, fatigue, feeling sick, hyperprolactinaemia and stomach pain. The results demonstrate that dizziness, weight gain, constipation, abdominal pain, backache, diarrhoea, hypotension, hyperprolactinemia, fatigue, and enuresis were more prevalent in females.

b. Ethnic background showed no significant associations with any ADRs.

c. Age group showed significant associations (See Fig 2) with agitation, fatigue, feeling sick, sedation, shaking, tachycardia and weight gain. The results show that agitation, sedation, dizziness, insomnia, convulsions, tachycardia and tremor were more prevalent in patients under 30 years of age. ADRs such as dry mouth, enuresis and hyperprolactinemia were prevalent in patients over 40 years of age, and dizziness, hypotension and hypertension were more prevalent in patients who are over 60 years old.

d. Hospital admission showed significant associations (See Fig 3) with abdominal pain, agitation, akathisia, backache, confusion, constipation, convulsion, diarrhoea, dizziness, dry mouth, dyspepsia, enuresis, fatigue, feeling sick, fever, headache, hyperprolactinemia, hypersalivation, hypertension, hypotension, insomnia, nausea, rash, sedation, shaking, stomach pain, sweating, tachycardia, tremor, vomiting and weight gain.

e. Smoking status showed associations (See Fig 4) with abdominal pain, agitation, backache, confusion, convulsion, diarrhoea, dizziness, dyspepsia, enuresis, fatigue, feeling sick, fever, headache, insomnia, sedation, shaking, stomach pain, sweating, tachycardia, vomiting and weight gain. The complete results are available in (S3 Table).

## Discussion

The study presents a medication continuity timeline (start and stop dates) for patients under Clozapine treatment to obtained detailed insight of Clozapine-induced ADRs using data from three large UK-based mental health Trusts comprising over 50 million documents and over half a million patients. The study uses the ADEPt NLP pipeline [56] to extract ADEs from free-text psychiatric EHRs, as well as a set of algorithms for creating a medication continuity timeline. The timeline was used to investigate associations between medications and ADEs, characterising ADR susceptibility with respect to patient demographics, hospital admission

**Table 2. The results are shown in percentages (%) and broken down by ADRs, Hospitals (SLAM, Camden & Islington and Oxford) and SIDER.**

| ADR | Trust | Three Months Early | Two Months Early | One Month Early | One Month Later | Two Months Later | Three Months Later | SIDER Low End | SIDER High End |
|---|---|---|---|---|---|---|---|---|---|
| Agitation | SLAM | 17.61 | 22.10 | 26.53 | 46.59 | 32.56 | 26.99 | | |
| | Camden & Islington | 13.37 | 17.83 | 18.36 | 43.14 | 28.34 | 21.03 | | |
| | Oxford (n = 514) | 14.59 | 15.76 | 16.34 | 34.24 | 25.10 | 20.62 | | |
| | SIDER | | | | | | | 4.00 | |
| Fatigue | SLAM | 12.67 | 14.83 | 15.85 | 43.58 | 35.80 | 30.51 | | |
| | Camden & Islington | 10.34 | 12.30 | 13.37 | 41.18 | 29.23 | 26.56 | | |
| | Oxford (n = 514) | 9.73 | 11.87 | 12.06 | 35.21 | 27.43 | 26.85 | | |
| | SIDER | | | | | | | | |
| Sedation | SLAM | 12.67 | 12.16 | 14.83 | 43.86 | 35.51 | 29.83 | | |
| | Camden & Islington | 5.17 | 9.09 | 9.09 | 38.15 | 26.56 | 21.93 | | |
| | Oxford (n = 514) | 7.20 | 8.37 | 9.34 | 31.52 | 21.40 | 18.48 | | |
| | SIDER | | | | | | | 25.00 | 46.00 |
| Dizziness | SLAM | 2.78 | 4.20 | 4.09 | 16.59 | 13.13 | 11.19 | | |
| | Camden & Islington | 3.21 | 3.39 | 3.74 | 18.18 | 13.73 | 9.09 | | |
| | Oxford (n = 514) | 3.89 | 4.09 | 4.47 | 17.70 | 13.04 | 10.12 | | |
| | SIDER | | | | | | | 12.00 | 27.00 |
| Hypersalivation | SLAM | 1.19 | 1.48 | 2.10 | 14.32 | 13.24 | 11.31 | | |
| | Camden & Islington | 1.07 | 1.43 | 0.53 | 14.26 | 6.95 | 7.66 | | |
| | Oxford (n = 514) | 0.97 | 0.78 | 1.56 | 12.65 | 10.70 | 5.84 | | |
| | SIDER | | | | | | | 1.00 | 48.00 |
| Feeling sick | SLAM | 4.66 | 4.94 | 6.48 | 14.32 | 11.19 | 9.09 | | |
| | Camden & Islington | 3.74 | 3.92 | 3.03 | 10.52 | 7.13 | 7.66 | | |
| | Oxford (n = 514) | 3.89 | 5.25 | 5.06 | 14.20 | 9.73 | 7.20 | | |
| | SIDER | | | | | | | | |
| Weight gain | SLAM | 3.75 | 4.43 | 5.06 | 15.34 | 10.91 | 10.34 | | |
| | Camden & Islington | 2.50 | 3.39 | 1.96 | 11.76 | 6.60 | 6.24 | | |
| | Oxford (n = 514) | 3.50 | 3.31 | 3.70 | 11.28 | 9.92 | 7.78 | | |
| | SIDER | | | | | | | 4.00 | 56.00 |
| Tachycardia | SLAM | 2.27 | 2.05 | 2.50 | 15.40 | 12.95 | 9.94 | | |
| | Camden & Islington | 1.43 | 1.43 | 0.89 | 11.23 | 8.38 | 6.95 | | |
| | Oxford (n = 514) | 0.78 | 1.36 | 1.56 | 10.89 | 10.51 | 7.59 | | |
| | SIDER | | | | | | | 11.00 | 25.00 |

*(Continued)*

**Table 2.** (Continued)

| ADR | Trust | Three Months Early | Two Months Early | One Month Early | One Month Later | Two Months Later | Three Months Later | SIDER Low End | SIDER High End |
|---|---|---|---|---|---|---|---|---|---|
| Confusion | SLAM | 4.72 | 5.51 | 6.08 | 13.92 | 8.47 | 6.76 | | |
| | Camden & Islington | 3.57 | 6.24 | 5.53 | 12.66 | 6.77 | 5.88 | | |
| | Oxford (n = 514) | 2.53 | 3.89 | 3.89 | 9.92 | 6.42 | 5.25 | | |
| | SIDER | | | | | | | 3.00 | |
| Constipation | SLAM | 1.76 | 1.99 | 2.16 | 12.27 | 11.70 | 9.49 | | |
| | Camden & Islington | 1.07 | 2.50 | 1.78 | 11.41 | 7.13 | 5.70 | | |
| | Oxford (n = 514) | 0.58 | 0.97 | 1.36 | 10.31 | 7.78 | 7.78 | | |
| | SIDER | | | | | | | 10.00 | 25.00 |
| Headache | SLAM | 4.20 | 4.55 | 5.45 | 12.44 | 8.18 | 5.91 | | |
| | Camden & Islington | 2.32 | 3.57 | 4.28 | 9.27 | 6.42 | 4.63 | | |
| | Oxford (n = 514) | 3.89 | 3.89 | 4.09 | 10.89 | 8.37 | 7.59 | | |
| | SIDER | | | | | | | | |
| Insomnia | SLAM | 3.92 | 4.03 | 5.17 | 10.40 | 6.48 | 4.03 | | |
| | Camden & Islington | 3.57 | 3.39 | 3.74 | 8.91 | 3.39 | 4.28 | | |
| | Oxford (n = 514) | 5.84 | 4.86 | 5.84 | 8.37 | 6.81 | 4.09 | | |
| | SIDER | | | | | | | 20.00 | 33.00 |
| Hyperprolactinaemia | SLAM | 3.18 | 3.64 | 4.20 | 8.52 | 5.06 | 4.15 | | |
| | Camden & Islington | 1.60 | 1.78 | 2.67 | 8.20 | 4.10 | 3.57 | | |
| | Oxford (n = 514) | 3.70 | 4.09 | 4.28 | 8.75 | 4.47 | 4.86 | | |
| | SIDER | | | | | | | | |
| Shaking | SLAM | 3.13 | 2.95 | 3.92 | 9.55 | 5.40 | 5.06 | | |
| | Camden & Islington | 1.78 | 1.96 | 3.74 | 6.06 | 3.92 | 2.85 | | |
| | Oxford (n = 514) | 2.92 | 3.31 | 3.89 | 7.78 | 4.47 | 4.86 | | |
| | SIDER | | | | | | | | |
| Vomiting | SLAM | 2.56 | 2.50 | 3.01 | 8.86 | 6.82 | 5.00 | | |
| | Camden & Islington | 2.14 | 2.50 | 2.85 | 6.77 | 4.99 | 4.63 | | |
| | Oxford (n = 514) | 1.75 | 2.72 | 2.92 | 7.59 | 5.25 | 5.06 | | |
| | SIDER | | | | | | | 3.00 | 17.00 |
| Hypertension | SLAM | 2.05 | 2.22 | 3.13 | 9.15 | 5.74 | 4.60 | | |
| | Camden & Islington | 0.71 | 0.71 | 1.60 | 7.13 | 4.63 | 2.67 | | |
| | Oxford (n = 514) | 1.36 | 1.36 | 1.56 | 5.06 | 4.28 | 2.14 | | |
| | SIDER | | | | | | | 4.00 | 12.00 |

(*Continued*)

**Table 2.** (Continued)

| ADR | Trust | Three Months Early | Two Months Early | One Month Early | One Month Later | Two Months Later | Three Months Later | SIDER Low End | SIDER High End |
|---|---|---|---|---|---|---|---|---|---|
| Abdominal pain | SLAM | 1.88 | 1.99 | 2.56 | 8.01 | 6.02 | 4.72 | | |
| | Camden & Islington | 0.89 | 0.89 | 1.60 | 3.92 | 3.57 | 3.39 | | |
| | Oxford (n = 514) | 1.75 | 1.36 | 1.75 | 7.39 | 4.47 | 5.64 | | |
| | SIDER | | | | | | | 4.00 | |
| Convulsion | SLAM | 1.36 | 1.70 | 1.82 | 7.05 | 4.94 | 4.03 | | |
| | Camden & Islington | 0.53 | 0.53 | 0.36 | 2.85 | 2.14 | 1.07 | | |
| | Oxford (n = 514) | 1.36 | 1.36 | 1.56 | 6.42 | 3.11 | 2.72 | | |
| | SIDER | | | | | | | 3.00 | |
| Backache | SLAM | 1.14 | 1.59 | 2.44 | 4.94 | 3.35 | 2.73 | | |
| | Camden & Islington | 1.43 | 1.25 | 1.96 | 5.35 | 3.03 | 3.03 | | |
| | Oxford (n = 514) | 1.17 | 1.17 | 1.36 | 5.84 | 3.89 | 2.92 | | |
| | SIDER | | | | | | | 5.00 | |
| Nausea | SLAM | 1.14 | 1.08 | 1.19 | 6.08 | 5.23 | 3.69 | | |
| | Camden & Islington | 0.89 | 1.43 | 0.36 | 4.63 | 3.57 | 3.57 | | |
| | Oxford (n = 514) | 0.97 | 0.58 | 1.36 | 4.86 | 3.70 | 2.92 | | |
| | SIDER | | | | | | | 3.00 | 17.00 |
| Hypotension | SLAM | 0.51 | 0.97 | 0.80 | 5.00 | 2.95 | 2.56 | | |
| | Camden & Islington | 0.18 | 0.53 | 0.18 | 3.57 | 2.32 | 1.78 | | |
| | Oxford (n = 514) | 0.58 | 0.78 | 0.78 | 5.64 | 3.50 | 2.92 | | |
| | SIDER | | | | | | | 9.00 | 38.00 |
| Fever | SLAM | 1.02 | 1.14 | 1.65 | 6.36 | 4.43 | 3.13 | | |
| | Camden & Islington | 0.89 | 0.89 | 0.53 | 3.74 | 2.67 | 0.89 | | |
| | Oxford (n = 514) | 0.39 | 0.78 | 0.58 | 3.11 | 2.72 | 2.33 | | |
| | SIDER | | | | | | | 4.00 | 13.00 |
| Enuresis | SLAM | 1.02 | 0.80 | 1.25 | 4.20 | 3.92 | 3.24 | | |
| | Camden & Islington | 0.71 | 1.07 | 1.07 | 4.10 | 1.43 | 1.25 | | |
| | Oxford (n = 514) | 1.36 | 0.58 | 1.36 | 4.86 | 4.47 | 3.50 | | |
| | SIDER | | | | | | | | |
| Dry mouth | SLAM | 1.08 | 1.53 | 1.65 | 4.66 | 3.69 | 2.33 | | |
| | Camden & Islington | 1.25 | 1.25 | 1.07 | 3.92 | 2.14 | 0.89 | | |
| | Oxford (n = 514) | 1.36 | 1.36 | 1.56 | 3.89 | 1.36 | 2.33 | | |
| | SIDER | | | | | | | 5.00 | 20.00 |

(*Continued*)

**Table 2.** (*Continued*)

| ADR | Trust | Three Months Early | Two Months Early | One Month Early | One Month Later | Two Months Later | Three Months Later | SIDER Low End | SIDER High End |
|---|---|---|---|---|---|---|---|---|---|
| Diarrhoea | SLAM | 1.08 | 1.31 | 1.36 | 4.72 | 3.58 | 2.56 | | |
| | Camden & Islington | 0.71 | 1.25 | 0.18 | 3.03 | 3.39 | 3.03 | | |
| | Oxford (n = 514) | 1.17 | 0.78 | 1.36 | 4.09 | 3.70 | 2.53 | | |
| | SIDER | | | | | | | 2.00 | |
| Rash | SLAM | 1.25 | 1.59 | 2.05 | 3.64 | 2.95 | 2.27 | | |
| | Camden & Islington | 1.25 | 1.25 | 0.89 | 4.28 | 1.96 | 2.14 | | |
| | Oxford (n = 514) | 0.97 | 1.17 | 1.17 | 3.70 | 2.33 | 1.36 | | |
| | SIDER | | | | | | | | |
| Dyspepsia | SLAM | 0.74 | 1.08 | 0.91 | 3.92 | 3.13 | 3.69 | | |
| | Camden & Islington | 0.36 | 0.53 | 0.53 | 4.10 | 2.67 | 2.50 | | |
| | Oxford (n = 514) | 0.19 | 0.58 | 0.78 | 3.50 | 4.09 | 3.70 | | |
| | SIDER | | | | | | | 8.00 | 14.00 |
| Stomach pain | SLAM | 1.93 | 1.76 | 1.93 | 4.94 | 3.52 | 3.52 | | |
| | Camden & Islington | 0.89 | 1.25 | 0.89 | 3.39 | 2.85 | 2.14 | | |
| | Oxford (n = 514) | 1.56 | 0.78 | 0.78 | 2.14 | 0.97 | 0.97 | | |
| | SIDER | | | | | | | | |
| Sweating | SLAM | 1.08 | 0.97 | 1.36 | 4.43 | 4.26 | 2.84 | | |
| | Camden & Islington | 0.53 | 0.53 | 0.53 | 2.85 | 2.14 | 1.96 | | |
| | Oxford (n = 514) | 1.17 | 0.97 | 0.97 | 2.72 | 1.36 | 1.95 | | |
| | SIDER | | | | | | | 6.00 | |
| Tremor | SLAM | 1.48 | 1.99 | 2.95 | 5.51 | 3.52 | 3.47 | | |
| | Camden & Islington | 1.60 | 1.78 | 2.14 | 3.92 | 1.96 | 2.14 | | |
| | SIDER | | | | | | | 6.00 | |
| Neutropenia | SLAM | 0.80 | 0.80 | 0.74 | 5.34 | 2.73 | 2.61 | | |
| | Camden & Islington | 0.00 | 0.18 | 0.53 | 1.60 | 0.89 | 1.07 | | |
| | SIDER | | | | | | | | |
| Akathisia | SLAM | 0.80 | 0.91 | 0.74 | 2.67 | 1.36 | 0.80 | | |
| | Camden & Islington | 0.00 | 0.53 | 0.00 | 1.25 | 1.07 | 0.53 | | |
| | Oxford (n = 514) | 0.97 | 0.78 | 0.97 | 1.36 | 1.17 | 0.97 | | |
| | SIDER | | | | | | | 3.00 | |
| Blurred vision | SLAM | 0.34 | 0.91 | 0.63 | 2.05 | 1.25 | 1.02 | | |
| | Camden & Islington | 0.89 | 0.53 | 0.71 | 1.25 | 0.36 | 0.89 | | |
| | Oxford (n = 514) | 0.19 | 0.39 | 0.39 | 1.56 | 1.56 | 1.17 | | |
| | SIDER | | | | | | | 5.00 | |

(*Continued*)

**Table 2.** (Continued)

| ADR | Trust | Three Months Early | Two Months Early | One Month Early | One Month Later | Two Months Later | Three Months Later | SIDER Low End | SIDER High End |
|---|---|---|---|---|---|---|---|---|---|
| | Legend | 0 | 10 | 20 | 30 | 40 | 50 | | |

The columns (Three Months Early, Two Months Early, One Month Early, One Month Later, Two Months Later, and Three Months Later) shows the percentages in each monthly interval. The last two columns (SIDER Low End and SIDER High End) shows the SIDER reporting.

and smoking status. Furthermore, the medication algorithm was used to assert the start date and measure the length of drug therapies. We used Clozapine as a use case, but the current work can be replicated on any psychotropic drug.

The algorithm found 2835 patients in the three trusts that started and continued Clozapine treatment for at least three months. The work presented in this paper has been run inclusively on all available ADEs in the ADEPt pipeline. We selected 33 ADRs to explore further as these were known side effects associated with Clozapine. We compared our results with SIDER, an online side effect resource. SIDER only reports on 25 out of the 33 ADRs. The 8 ADRs not reported in SIDER are fatigue, feeling sick, headache, hyperprolactinemia, weight loss, shaking, rash and stomach pain. These 8 ADRs show interesting results across the three trusts, as shown in Table 2. The prevalence of ADRs was assessed over gender, ethnic background, age groups, smoking status and hospital admission.

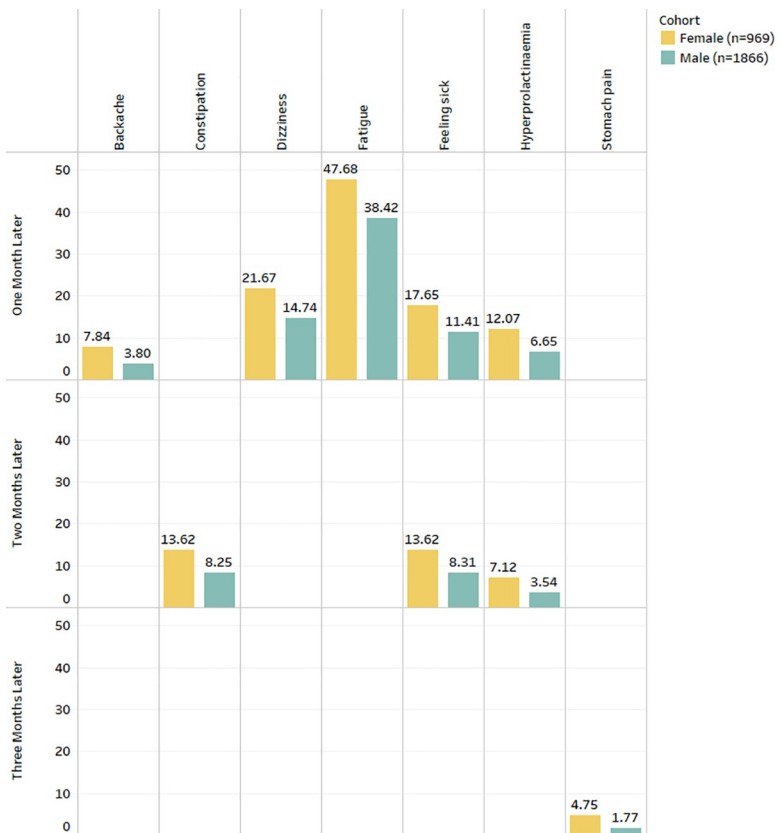

**Fig 1. Frequency distribution of statistically significant ADRs (after Bonferroni p-value adjustment) from the combined analysis in gender for three months after starting the drug Clozapine.**

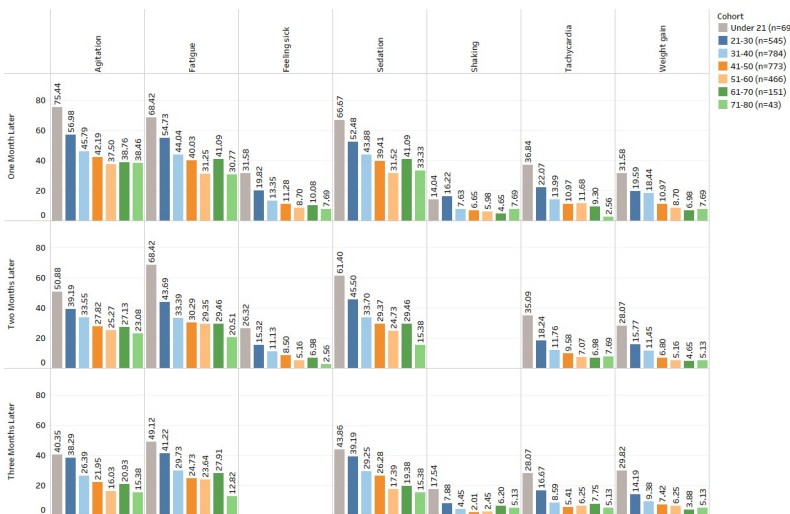

**Fig 2. Frequency distribution of statistically significant ADRs (after Bonferroni p-value adjustment) from the combined analysis in age groups for three months after starting the drug Clozapine.**

We stratified the data using demographics information, mainly gender, ethnic background, age groups, hospital admission and smoking status. Sedation, fatigue, agitation, hypersalivation, tachycardia, constipation, dizziness and weight gain are the most common [68–72] and most highly-recorded ADRs in all three NHS trusts. When comparing hospital admission data (inpatient vs outpatient), our results show that the inpatient group have a higher recording of any ADRs when the patient starts Clozapine therapy compared to the outpatient group. This is due to the inpatient group being more frequently monitored and recorded by clinicians. A similar pattern is observed when comparing smoking status (smoker vs non-smoker).

Rare ADRs such as agranulocytosis, myocarditis, SJS, cardiomyopathy and pericarditis are reported in the analysis. However, rare ADRs require much attention before they can be declared as positive. Currently, we are manually validating our results on agranulocytosis,

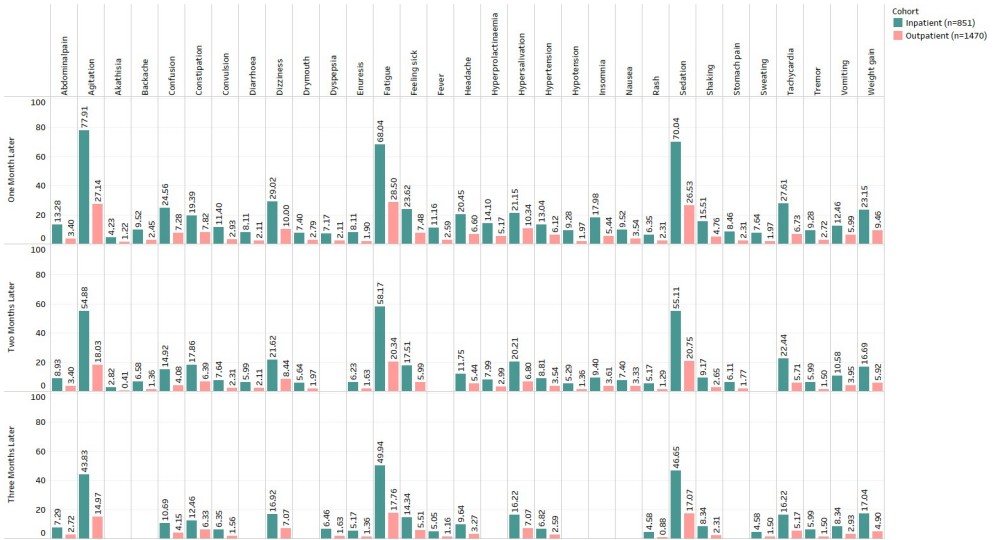

**Fig 3. Frequency distribution of statistically significant ADRs (after Bonferroni p-value adjustment) from the combined analysis in hospital admission for three months after starting the drug Clozapine.**

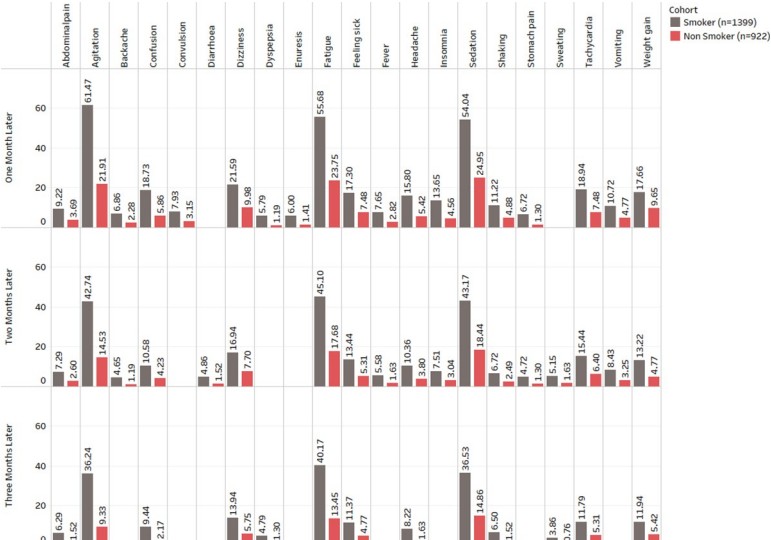

**Fig 4. Frequency distribution of statistically significant ADRs (after Bonferroni p-value adjustment) from the combined analysis in smoking status for three months after starting the drug Clozapine.**

myocarditis and neutropenia in parallel studies by going through the clinical notes to improve our assertions.

We also acknowledge that the results still may be specific to the UK in some way. SLAM and C&I trusts are based in London, the population are from very mixed backgrounds, and ethnicities and Oxford trust population are more homogenous. Therefore we believe the results are specific to the UK but may not cover all UK specific concerns.

## Limitation

This study was first conducted and evaluated using SLAM's psychiatric clinical notes. In SLAM, evaluation of each step was carried out manually by at least two annotators, and an inter-annotator agreement was achieved where possible. Due to limited resources, several challenges were faced when implementing the work in C&I and Oxford Trusts. The ADEPt pipeline and ADR timeline were not manually validated in Oxford Trust and the results presented in the C&I Trust are based on a single annotator. Although the proposed work has shown good results, lower precision and recall were found in rare ADEs as they are frequently recorded as warnings, potential and suspected occurrences.

## Future work

Ongoing work focuses on extending the analyses to longer periods of time, other drugs such as Lurasidone, drug-drug interaction of Clozapine, adding more patient-related features such as Body Mass Index (BMI), education, mobility, employment status, welfare benefits, homelessness, blood results and alcohol use. We are also planning to apply for this work on other drugs. Current work is underway on the antipsychotic Lurasidone and drug-drug interaction of Clozapine.

## Conclusion

ADR extraction from the free-text clinical documents can help clinicians and researchers to predict risks and interventions of drug administration. Often under-utilised due to the fatal

nature of ADR, Clozapine is the most effective drug for treatment-resistant schizophrenia. A number of limited studies report Clozapine-induced ADRs. One of its kind in terms of cohort size and the variety of ADRs, this study characterises and provides insight into Clozapine-induced ADRs on a large population of patients across three large mental health hospitals. As well as providing novel findings, the proposed method demonstrates the utility of wider ADR extraction beyond Clozapine, and the study can be replicated using any psychotropic drug. In the future, this work will be expanded to define extended periods of on-treatment episodes, other drugs and more patient-related features in the statistical analysis.

## Supporting information

**S1 Table. Clozapine–gender differences (%).**
(PDF)

**S2 Table. Chi square statistics ($\chi2$).**
(PDF)

**S3 Table. Combine analysis.**
(PDF)

## Acknowledgments

**Disclaimer:** The views expressed are those of the authors and not necessarily those of the NHS, the NIHR or the Department of Health.

## Author Contributions

**Conceptualization:** Ehtesham Iqbal, Olubanke Dzahini, Richard J. B. Dobson, Zina M. Ibrahim.

**Data curation:** Ehtesham Iqbal, Alvin Romero, Olubanke Dzahini, Chi-Hun Kim, Nomi Werbeloff, James H. MacCabe, Zina M. Ibrahim.

**Formal analysis:** Ehtesham Iqbal, Zina M. Ibrahim.

**Funding acquisition:** Tanya Smith, Richard J. B. Dobson.

**Investigation:** Ehtesham Iqbal, Robert Stewart, Chi-Hun Kim, Nomi Werbeloff, James H. MacCabe, Zina M. Ibrahim.

**Methodology:** Ehtesham Iqbal, Zina M. Ibrahim.

**Project administration:** Ehtesham Iqbal, Richard J. B. Dobson.

**Resources:** Matthew Broadbent, Robert Stewart, Tanya Smith, Chi-Hun Kim, Nomi Werbeloff.

**Software:** Ehtesham Iqbal, Matthew Broadbent, Tanya Smith, Nomi Werbeloff.

**Supervision:** Olubanke Dzahini, Tanya Smith, Nomi Werbeloff, Richard J. B. Dobson, Zina M. Ibrahim.

**Validation:** Ehtesham Iqbal, Risha Govind, Alvin Romero, Olubanke Dzahini, Chi-Hun Kim.

**Visualization:** Ehtesham Iqbal, Richard J. B. Dobson, Zina M. Ibrahim.

**Writing – original draft:** Ehtesham Iqbal, Richard J. B. Dobson, Zina M. Ibrahim.

**Writing – review & editing:** Ehtesham Iqbal, Risha Govind, Richard J. B. Dobson, Zina M. Ibrahim.

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
