## [Decision Letter · Decision Letter 0]

29 Jun 2020

PONE-D-20-04570

The side effect profile of Clozapine in real world data of three large mental health hospitals

PLOS ONE

Dear Dr. Iqbal,

Thank you for submitting your manuscript to PLOS ONE. After careful consideration, we feel that it has merit but does not fully meet PLOS ONE’s publication criteria as it currently stands. Therefore, we invite you to submit a revised version of the manuscript that addresses the points raised during the review process.

We look forward to receiving your revised manuscript.

Kind regards,

Vincenzo De Luca

Academic Editor

PLOS ONE

Journal Requirements:

2. Please note that PLOS ONE has guidelines on software sharing (https://journals.plos.org/plosone/s/materials-and-software-sharing#loc-sharing-software).

Accordingly, we encourage you to make the code for the algorithm described in your manuscript publicly available, if it has not been published in full previously.

Reviewers' comments:

Reviewer's Responses to Questions

**Comments to the Author**

1. Is the manuscript technically sound, and do the data support the conclusions?

Reviewer #1: Partly

2. Has the statistical analysis been performed appropriately and rigorously? 

Reviewer #1: Yes

3. Have the authors made all data underlying the findings in their manuscript fully available?

Reviewer #1: Yes

4. Is the manuscript presented in an intelligible fashion and written in standard English?

Reviewer #1: Yes

5. Review Comments to the Author

Reviewer #1: Iqbal et al. have undertaken the evaluation of side effect profile of clozapine using patient notes from three different hospitals in the UK.

Such endeavors are typically implemented via reaching large enough samples size. In addition, data should be generalizable, and therefore replication studies in an independent cohort are usually conducted to ensure clinical value. Replication also serves as a test of reproducibility of obtained findings. In the current study, authors used de-identified Electronic Health Records and included notes from 2,835 patients admitted into three mental health trusts in the UK. Side effects of clozapine were monitored for 3 months after the initiation of the medication. It was shown that most common side effects during 3-month period were relatively non-specific: sedation, fatigue, agitation, dizziness, hypersalivation, weight gain, tachycardia, headache, constipation, and confusion. To search for discrepancy between obtained findings and established side effect profile of clozapine, authors compared their data with the SIDER database. For most side effects, numbers were more or less similar, but several unexpected results appear to stand out (described in detail in a following paragraph). Statistics seems to be performed appropriately.

Regarding significance, clozapine is a last resort antipsychotic commonly used in patients with refractory SCZ who failed 3 trials with different antipsychotics. Clozapine is notorious for its side effect profile. Among most feared side effects are agranulocytosis, weight gain, seizures, myocarditis, and QT prolongation, all of which requiring careful monitoring. Even though screening of patient notes is generally deemed inferior to established follow-ups on clinical trials such as FAERS etc. (in part due to lack of consistency and absence of clearly set evaluation criteria), blatant discrepancies between established numbers and numbers in routine clinical practice are typically visible. Such discrepancies might be of clinical significance if, for instance, there are new side effects or frequency of established adverse reactions is different, as it may suggest additional or different monitoring strategies. Authors obtained some interesting findings. For example, it was shown that weight gain occurred mostly during the first month, a phenomenon which may deserve further investigation. Incidence of insomnia appeared to be less than previously thought. Furthermore, number of patients who complained of dry mouth and/or blurred vision was less suggesting that anticholinergic effects of clozapine could have been overestimated. Authors also detected higher incidence of diarrhea which is also suggestive of relatively weak anticholinergic activity of clozapine. Of note, agitation was shown to be more prevalent than previously shown. Less incidence of hypotension and dyspepsia may represent notable findings as well. Also, more precise numbers were obtained for hypersalivation and weight gain. Some of these data may have a potential to shift the emphasis in monitoring. Reviewer would select the weight gain (which was shown to be most prominent during first month) and higher incidence of agitation as deserving a follow-up investigation. Different numbers were also obtained for confusion, but confusion is somewhat subjective criterion, so that these data should be interpreted with caution.

Reviewer has several concerns, though. The biggest concern is the lack of a replication cohort. As mentioned in a previous paragraph, in case of confusion as well as other subjective side effects (such as insomnia and dyspepsia), there might be a lot of discrepancies between departments and hospitals, as criteria for subjective side effects may differ among providers and specialties. Furthermore, there is a possibility of Berkson bias, as it is unclear to what extent data obtained in the UK are generalizable. It has been known that genetic architecture plays an important role in pathogenesis of psychotic disorders as well as in pharmacodynamics of clozapine, and it is not unlikely that obtained data may not be generalizable and/or replicated. Second, authors may want to clarify why 3-month period was selected for observation. Among most clinically significant side effects are agranulocytosis, myocarditis, and seizures, and these effects may take longer to develop.

There are also few minor comments. To what extend it is OK to use SIDER database as the only reference? The latest version of SIDER (4.1) was released in 2015, and data on side effects of clozapine listed there might be somewhat outdated. Could results be compared to an additional database? Furthermore, Table 2 appears to be redundant, numbers from this table likely could be given in the text in parentheses. Authors also may want to describe statistics in a more detail, especially how adjustments to potential confounders were made. Figures 1 and 2 seem to be redundant, and authors may consider removing them.

6. PLOS authors have the option to publish the peer review history of their article (what does this mean?). If published, this will include your full peer review and any attached files.

Reviewer #1: No

---

## [Author Response · Author response to Decision Letter 0]

16 Nov 2020

Dear Reviewer, 

We hope that we have addressed all of the reviewer's’ concerns and feel that the paper has been substantially improved. Please see the attached copy of manuscript labeled ‘Revised manuscript with Track Changes’ and revised paper without track changes.

Kind Regards

Ehtesham Iqbal

---

## [Decision Letter · Decision Letter 1]

23 Nov 2020

The side effect profile of Clozapine in real world data of three large mental health hospitals

PONE-D-20-04570R1

Dear Dr. Iqbal,

We’re pleased to inform you that your manuscript has been judged scientifically suitable for publication and will be formally accepted for publication once it meets all outstanding technical requirements.

Kind regards,

Vincenzo De Luca

Academic Editor

PLOS ONE

Additional Editor Comments (optional):

Reviewers' comments:

Reviewer's Responses to Questions

**Comments to the Author**

1. If the authors have adequately addressed your comments raised in a previous round of review and you feel that this manuscript is now acceptable for publication, you may indicate that here to bypass the “Comments to the Author” section, enter your conflict of interest statement in the “Confidential to Editor” section, and submit your "Accept" recommendation.

Reviewer #1: All comments have been addressed

2. Is the manuscript technically sound, and do the data support the conclusions?

Reviewer #1: Yes

3. Has the statistical analysis been performed appropriately and rigorously? 

Reviewer #1: Yes

4. Have the authors made all data underlying the findings in their manuscript fully available?

Reviewer #1: Yes

5. Is the manuscript presented in an intelligible fashion and written in standard English?

Reviewer #1: Yes

6. Review Comments to the Author

Reviewer #1: (No Response)

7. PLOS authors have the option to publish the peer review history of their article (what does this mean?). If published, this will include your full peer review and any attached files.

Reviewer #1: No

---

## [Editor Report · Acceptance letter]

27 Nov 2020

PONE-D-20-04570R1 

The side effect profile of Clozapine in real world data of three large mental health hospitals 

Dear Dr. Iqbal:

I'm pleased to inform you that your manuscript has been deemed suitable for publication in PLOS ONE. Congratulations! Your manuscript is now with our production department. 

Kind regards, 

on behalf of

Dr. Vincenzo De Luca 

Academic Editor

PLOS ONE